# Interplay between Cellular Metabolism and the DNA Damage Response in Cancer

**DOI:** 10.3390/cancers12082051

**Published:** 2020-07-25

**Authors:** Amandine Moretton, Joanna I. Loizou

**Affiliations:** 1CeMM Research Center for Molecular Medicine of the Austrian Academy of Sciences, 1090 Vienna, Austria; 2Institute of Cancer Research, Department of Medicine I, Medical University of Vienna and Comprehensive Cancer Center, 1090 Vienna, Austria

**Keywords:** DNA repair, DNA damage, DNA damage response, metabolism, high-throughput technologies

## Abstract

Metabolism is a fundamental cellular process that can become harmful for cells by leading to DNA damage, for instance by an increase in oxidative stress or through the generation of toxic byproducts. To deal with such insults, cells have evolved sophisticated DNA damage response (DDR) pathways that allow for the maintenance of genome integrity. Recent years have seen remarkable progress in our understanding of the diverse DDR mechanisms, and, through such work, it has emerged that cellular metabolic regulation not only generates DNA damage but also impacts on DNA repair. Cancer cells show an alteration of the DDR coupled with modifications in cellular metabolism, further emphasizing links between these two fundamental processes. Taken together, these compelling findings indicate that metabolic enzymes and metabolites represent a key group of factors within the DDR. Here, we will compile the current knowledge on the dynamic interplay between metabolic factors and the DDR, with a specific focus on cancer. We will also discuss how recently developed high-throughput technologies allow for the identification of novel crosstalk between the DDR and metabolism, which is of crucial importance to better design efficient cancer treatments.

## 1. Introduction

Maintaining genome integrity via repair of DNA damage is a key biological process required to suppress diseases. In response to DNA damage, cells elicit a signaling cascade resulting in the activation of specific repair machinery, depending on the type of damage. The DNA damage response (DDR) has been well-studied in recent years, broadening our knowledge of the diverse DDR pathways. Thereby, it has emerged that cellular metabolism is tightly connected to DNA repair, and can also directly lead to the generation of DNA damage [1]. Cancer cells present both an extensive metabolic reprogramming and alterations in DNA repair pathways, emphasizing the links between metabolism and the DDR. The Warburg effect is one of the main metabolic changes during carcinogenesis and consists of an increased use of anaerobic glycolysis, rather than oxidative phosphorylation, to sustain proliferation. In this review, we will compile the current knowledge on the interplay between metabolic factors and the DDR, with a specific focus on cancer (Figure 1). Also, we will present high-throughput technologies and approaches that allow us to further unravel the regulation of DDR mechanisms by metabolism.

## 2. Oxidative Stress and the DNA Damage Response

### 2.1. Oxidative DNA Damage is Induced by Metabolic Reactions

Metabolism is linked to the DNA damage response through redox homeostasis. Reactive oxygen species (ROS) produced during metabolism include the free radicals superoxide (O_2_) and hydroxyl (OH), as well as hydrogen peroxide (H_2_O_2_). Among the different cellular sources of ROS (reviewed in [2]), the electron transport chain is the main producer of superoxide [3]. Besides their production by metabolic reactions, free radicals can also be generated during inflammatory processes, including virus exposure, by activated macrophages and neutrophils [4]. Reaction of ROS with DNA mainly occurs through reaction of OH with purines, pyrimidines or sugars of the DNA backbone, and one of the most frequent oxidative DNA lesion is 8-oxo-dG [5]. Base excision repair (BER) is the main pathway that removes oxidized bases. Incomplete repair generates abasic sites and single-strand DNA breaks (SSBs) as promutagenic repair intermediates. The accumulation of two or more DNA damage sites, such as oxidized bases, abasic sites and SSBs, on opposing strands within a few helical turns, is known as oxidatively generated clustered DNA lesions (OCDLs). OCDLs have been shown to occur not only after ionizing radiation (IR), but also endogenously. Closely spaced lesions induced by oxidative stress can generate DSBs, thus posing an important threat to genome stability [6,7]. Furthermore, unrepaired oxidized bases can lead to base mispairing during replication. Hence, oxidative stress strongly impacts on genome integrity. ROS can additionally create oxidative damage on other biomolecules, such as proteins and lipids. Lipid peroxidation specifically is an important endogenous source of DNA damage (see Section 3). However, it should be recognized that, while ROS production can be detrimental to cells, it can also be beneficial since a limited amount of free radicals is required to stimulate cellular responses and upregulate antioxidant pathways [8].

The oxidative stress hypothesis is one of the prevailing theories of aging. This theory postulates that metabolic reactions produce free radicals that in turn damage biomolecules, and that this accumulation of oxidative damage drives the aging process. A few decades ago, a study in four different mammalian species showed that metabolic rates and excretion of oxidized DNA bases in urine are inversely correlated with lifespan, thus supporting this hypothesis [9]. Moreover, it has been observed that larger mammals, such as elephants or whales, despite possessing more cells and a longer lifetime which implies more exposure to mutagens, have a lower incidence of cancers compared to smaller animals [10]. This phenomenon is known as Peto’s paradox and can be explained by several hypotheses. According to one hypothesis, larger and long-lived organisms have evolved highly regulated tumor suppressor mechanisms. For example, elephants have twenty copies of the tumor suppressor *TP53*, while humans and most other mammals have only one [11,12]. Furthermore, larger animals have larger cells which divide slower, thus reducing the accumulation of mutations that occur during DNA replication [13]. This resistance to cancer initiation has also been hypothesized to be due to slower metabolic rates. As such, larger mammals require less energy to maintain body temperature—due to a reduced body surface area to body mass ratio—and are usually rather sluggish animals [14]. Thus, their slower metabolic rate leads to the generation of fewer free radicals and oxidized DNA damage and, therefore, less mutations that might contribute to the initiation of cancer. Along the same line, reduction in daily caloric intake and periods of fasting can also increase longevity and prevent or postpone the onset of chronic diseases in mammalian models [15,16]. These dietary habits have an impact on oxidative stress and can prevent the initiation and progression of cancers [17,18,19]. In support of this, metformin, a drug commonly used for type II diabetes, is associated with a substantial reduction in cancer risk according to pharmacoepidemiologic studies. This preventive effect can be explained by interference of the drug with the respiratory complex I and resulting inhibition of oxidative phosphorylation, associated with reduced endogenous ROS production and oxidative stress, therefore leading to reduced DNA damage and mutagenesis [20]. Additionally, obesity is a known risk factor for multiple types of cancers [21]. First, obese individuals present increased ROS production, due to activation of NOX4, the main nicotinamide adenine dinucleotide phosphate (NADPH) oxidase isoform in adipocytes and enhanced mitochondrial ß-oxidation following accumulation of triglycerides. Accumulation of fat in adipocytes also causes an infiltration of T-lymphocytes and macrophages into adipose tissue, promoting ROS production by NOX2, the NADPH oxidase expressed in inflammatory cells. The secretion of cytokines by activated macrophages additionally generates oxidative stress. Obesity-induced inflammation is thus involved in oxidative DNA damage and cancer initiation [22]. Moreover, since cancer cells have a high metabolic rate in order to sustain proliferation, they are prone to accumulation of oxidative DNA damage, which can further drive tumorigenesis [23,24].

### 2.2. Metabolism Functions in the Antioxidant Response

Intracellular levels of ROS reflect a balance between their generation and detoxification. Due to increased oxidative stress, cancer cells strongly rely on antioxidant signaling for survival [25,26]. ROS detoxification mainly involves the ROS scavenger glutathione (GSH), which acts as a reducing agent. Oxidized glutathione (GSSG) is regenerated by glutathione reductase using NADPH as a co-factor. Intracellular levels of GSH and NADPH, tightly regulated by metabolic reactions and often altered in cancers, are thus of crucial importance for redox homeostasis and limitation of DNA damage [27,28].

NADPH is produced from its oxidized form, NADP^+^, by glucose-6-phosphate dehydrogenase (G6PD) in the first step of the pentose phosphate pathway (PPP). Mouse models with a modest increase in G6PD expression have higher levels of NADPH, which correlates with decreased levels of oxidative DNA damage and improved health span [29]. Similarly, ATM, a serine/threonine protein kinase involved in the activation of the DNA damage checkpoint, promotes Heat Shock Protein 27 (Hsp27) phosphorylation and binding to G6PD, which in turn enhances G6PD activity, and thus triggers an antioxidant response [30]. Furthermore, ATM is activated as a disulfide-crosslinked dimer upon a low level of ROS, independently from MRN (Mre11, Rad50, and Nbs1)-dependent activation, where ATM undergoes a dimer-to-monomer transition. Hence, it regulates the global cellular response to oxidative stress independently from DNA damage [31]. Surprisingly, disruption of the PPP can also protect cells against ROS. Phosphogluconate dehydrogenase (PGD) depletion leads to accumulation of the upstream metabolite 6-phosphogluconate (6PG), which inhibits glycolysis. Therefore, glucose flux through the PPP is increased and this metabolic rewiring enhances NADPH production [32]. The synthesis of NADPH also depends on NAD^+^ production, since it is utilized to synthesize NADP^+^. NAD^+^ is produced by several redundant pathways, involving two key enzymes: nicotinate phosphoribosyltransferase (NAPRT) and nicotinamide phosphoribosyltransferase (NAMPT) [33]. Cancer cells have an accelerated rate of NAD^+^ cycling relative to untransformed cells. They have a specific dependency on either NAMPT or NAPRT, depending on the cancer type [34], and cells treated with inhibitors of either of these two enzymes show enhanced susceptibility to oxidative stress [35,36,37]. Untransformed cells are less sensitive to ROS induction upon NAMPT inhibition, suggesting a therapeutic window that can be used to target a specific vulnerability in cancer cells [38,39]. Moreover, cytotoxicity induced by glucose deprivation is enhanced in cancer cells and is mediated by increased steady-state levels of ROS [40]. This supports the hypothesis that metabolic reprogramming towards glycolysis and the PPP in cancer is necessary to provide antioxidants for ROS detoxification. Therefore, glucose deprivation induces NAMPT, which protects tumor cells from cell death that is caused by glucose starvation-induced ROS stress [37].

Besides regeneration of the oxidized glutathione using NADPH, GSH synthesis also involves other metabolic pathways and metabolites. S-adenosylmethionine (SAM) is one of the precursors of GSH and, through its conversion into S-adenosylhomocysteine, can be further hydrolyzed into homocysteine. Homocysteine can react with serine and be converted in two steps into cysteine, a key metabolite in GSH synthesis. Instead of being converted into cysteine, homocysteine can be irreversibly remethylated into methionine, but high SAM levels inhibit this reaction, therefore stimulating cysteine synthesis and GSH production [41]. Hence, SAM increases ROS detoxification through glutathione metabolism modulation [42,43].

### 2.3. DNA Repair Following Oxidative Stress Depends on the Metabolite NAD^+^

As well as being a precursor of NADP^+^ and NADPH, NAD^+^ is also an essential substrate for different DNA repair enzymes, in particular poly-ADP-ribose polymerases (PARPs) and sirtuins. One of the earliest events in the DDR is the recruitment of PARP1 to various DNA lesions, where it signals the damage and activates repair factors by poly(ADP)ribosylation (PARylation), which consists of the polymerization of ADP-ribose units derived from the ADP donor NAD^+^. PARP enzymes play a crucial role in regulating the cellular responses to oxidative stress. Indeed, PARP1/2 favor cell survival under mild oxidative stress, while extensive PARylation due to high levels of ROS and DNA damage leads to cell death to prevent accumulation of mutations [44]. Following DNA damage, p53 is PARylated by PARP1, inhibiting its nuclear export and thus promoting its transactivation activities, such as cell cycle arrest [45]. Moreover, in response to oxidative stress, PARP1 interacts with the 20S proteasome, stimulating its proteolytic activity and the removal of oxidized proteins. For example, nucleosomal histones protect DNA from free radical-mediated damage, and PARP1 enhances the selective degradation of oxidatively damaged histones [46]. Inhibitors of NAMPT and NAPRT, which decrease NAD^+^ levels, also affect PARP activity and thus can inhibit the DNA repair capacities of tumors. Furthermore, dual inhibition of PARP and NAMPT is synthetic lethal in Ewing Sarcoma and triple-negative breast cancer cells [47,48]. Similarly, NADP^+^ is an endogenous inhibitor of PARP and impairs PARylation-dependent DNA repair. Therefore, NADP^+^ levels might be used as a predictive biomarker for therapeutic potential of PARP inhibitors in cancer treatment [49].

Sirtuins are NAD^+^-dependent deacetylases involved both in the DDR and in metabolism. They play an important role in cancer biology, for example by repressing p53 by deacetylation, thus preventing cell death of tumor cells, which can be reversed by NAMPT inhibition [50,51,52]. Therefore, they have an opposing effect on p53 compared to PARP1. Additionally, SIRT1 is involved in oxidative stress-dependent cell death, through its interaction with the checkpoint kinase 2 (CHK2). CHK2 is activated by the ATM kinase in response to genotoxic stress including ROS to induce cell death or G2/M arrest. In normal conditions CHK2 is inhibited by SIRT1 via deacetylation, but, in the presence of elevated oxidative stress, CHK2 dissociates from SIRT1 and becomes activated by acetylation, thus leading to cell death [53]. Finally, sirtuins and PARPs have been shown to function together in the DDR following oxidative stress, thus connecting stress signaling and DNA repair pathways. They notably compete for the same substrate, NAD^+^, and therefore overactivation of PARP1 upon excessive DNA damage leads to repression of SIRT1 activity. Conversely, a reduction in PARP1 activity is observed following SIRT1 activation, due to PARP1 deacetylation, which blocks its activity [54]. In summary, interconnections between cellular metabolism and the DDR in oxidative stress are manifold. The detoxification of ROS generated by metabolic reactions is crucial to avoid oxidative DNA damage and an excessive burden on the DNA repair machinery, and yet both the synthesis of antioxidants and the DDR depend on metabolism to function.

## 3. DNA Adducts Are Produced through Metabolic Reactions

DNA adducts pose an important threat to genome integrity. They can either result in mutations, since many reactive compounds bind to the amino groups participating in Watson–Crick base pairing, or block DNA replication. Metabolism is a major source of DNA adducts. Firstly, many food carcinogens have been identified, mainly in processed or cooked food [55,56]. Secondly, toxic metabolites are endogenously produced by metabolic reactions that are not only linked to diet. This review will focus on this second type of metabolites generating DNA adducts, and more specifically on the most-studied aldehydes and alkylating agents (Table 1).

Different repair pathways are involved in protecting genome integrity following DNA adduct generation: base adducts are removed by BER or nucleotide excision repair (NER) in the case of bulky adducts. Intrastrand crosslinks are resolved by NER, while interstrand crosslinks are repaired by the Fanconi Anemia pathway and homologous recombination (HR). Finally, DNA–protein crosslinks are mainly repaired by HR. However, repair of DNA adducts can itself cause mutations, because intermediary steps generate abasic sites or strand breaks, which, if the repair process is not completed in a timely fashion, can lead to mutations [121,122,123]. Since DNA adducts cause genome instability, they result in loss of normal growth-control mechanisms and hence promote carcinogenesis.

### 3.1. Aldehydes

Formaldehyde, whose systematic name is methanal, is the simplest of the aldehydes and is highly toxic to cells. Endogenously, it is produced by enzymatic oxidative demethylation reactions, methylamine metabolism, and myeloperoxidation [64,65,66]. It is also a byproduct of methanol metabolism [67]. Formaldehyde reacts with guanine, adenine, and cytosine to form hydroxymethyl adducts [61], which then cause base substitutions during replication. Moreover, the formation of DNA intrastrand crosslinks is responsible for tandem base substitutions [124]. Interstrand crosslinks and DNA–protein crosslinks can also cause DNA replication blocks, DNA strand breaks, and deletions.

Acetaldehyde is another well-studied endogenous aldehyde. It is produced during ethanol metabolism and from pyruvate, threonine, and other metabolic reactions [73,74,125]. The major lesion caused by acetaldehyde is N^2^-ethylidene-deoxyguanosine, which can in turn become stabilized by reduction into N^2^-ethyl-deoxyguanosine [69,70]. This adduct is thought to be mainly bypassed by DNA polymerases in a nonmutagenic manner [126,127] but can also cause frameshift deletions [128]. The genotoxicity of acetaldehyde presumably comes from the generation of intrastrand, interstrand, and DNA–protein crosslinks [129].

Next, methylglyoxal is a carcinogenic metabolite produced as a side-product during glycolysis and degradation of acetone, aminoacetone, threonine, and glycated proteins [80,81,82,130]. It is a major reactive carbonyl species (RCS). Methylglyoxal-derived adducts on DNA are mainly deoxyguanosine-derived imidazopurinones of DNA, which can lead to depurination and promutagenic abasic sites [75,76]. Interstrand or DNA–protein crosslinks can also be formed and are highly toxic lesions, resulting in replication blocks and deletions, for example due to collapse of replication-forks. Crosslinks between the modified DNA template and the Klenow fragment of the DNA polymerase are also generated [79]. Moreover, methylglyoxal readily reacts with proteins and, notably, histones. Histone glycation affects chromatin compaction and disrupts chromatin architecture, thus impacting DNA repair. This histone modification also leads to toxic histone–DNA crosslinks [77].

Finally, α,β-unsaturated aldehydes are another class of metabolites that are reactive towards DNA. They are mainly produced endogenously through lipid peroxidation, arising as a consequence of oxidative stress [60]. Unlike reactive free radicals, they are stable enough to diffuse from their site of origin. As such, they react with DNA nucleobases to give exocyclic adducts that are responsible for blocking the Watson–Crick base pairing region, and thus lead to mispairing during replication. Among them, malondialdehyde (MDA) appears to be the most mutagenic. It forms adducts on adenosine, cytosine, and guanosine, with the latter occurring five times more frequently [131]. Mutations resulting from these DNA modifications are mainly base substitutions and large indels that occur due to interstrand crosslinks as premutagenic lesions. Other α,β-unsaturated aldehydes are 2,3-epoxy-4-hydroxynonanal (HNE), acrolein, and crotonaldehyde. These form propano adducts on deoxyguanosine, which then principally lead to transversions [60]. Interstrand and DNA–protein crosslinks are other consequences of these adducts. Furthermore, α,β-unsaturated aldehydes can be converted to epoxyaldehydes through different oxidative processes and will then form etheno adducts on DNA. Since they are more reactive than the parental enals, they give rise to adducts on deoxyguanosine, deoxycytidine, and deoxyadenosine, hence inducing replication blocks that are bypassed by error-prone translesion synthesis polymerases [98].

In addition to their roles in generating DNA damage, these aldehydes have been shown to also influence DNA repair. First, they compromise BER and NER pathways, therefore contributing to their procarcinogenic effects [132,133,134,135]. Indeed, high concentrations of HNE, for example, increase the rate of abasic site incision and subsequently block the re-ligation step after gap-filling by DNA polymerases, thus increasing the number of SSBs in cells treated with oxidizing or methylating agents. However, the mechanism by which it modulates BER enzyme activities remains unknown. In addition, HNE, which is a substrate of aldehyde dehydrogenase 2 (ALDH2), reduces its catalytic activity by forming an adduct with a cysteine of the catalytic site, and therefore a high concentration of HNE prevents its correct clearance [136]. Moreover, it has been shown that the preferential adduct site of acrolein and HNE to *TP53* coincides with a mutational hotspot in some cancer [135,137]. Mutations in *TP53*, a tumor suppressor involved in the DDR, are an early event in carcinogenesis, which further confirms that reactive metabolites are tumorigenic.

### 3.2. Alkylating Agents

Alkylating agents are the second main class of compounds which generate mutagenic DNA adducts. The major source of endogenous methylation is SAM, produced by the reaction of ATP with methionine and a key compound in transmethylation, aminopropylation and transsulfuration [113]. Methylation of DNA is an essential process in epigenetic regulation, and it functions to repress gene expression through methylation of gene promoters. However, SAM can also react nonenzymatically with DNA, creating mutagenic adducts. Both nitrogen and oxygen atoms of DNA bases can be alkylated, but SAM forms preferentially N7-methylguanine (7meG), as well as N1- and N3-methyladenine (1meA and 3meA) [110]. 7meG is relatively harmless but its destabilization can lead to depurination and a mutagenic abasic site. On the contrary, the 1meA and 3meA modifications block DNA replication, but not the activity of promutagenic translesion synthesis polymerases. Adducts on oxygen atoms from guanines and thymines are less prevalent but highly mutagenic, leading to mispairing during DNA replication [114].

Ethylene oxide (EO), produced from the metabolism of ethylene in the liver, is another potent alkylating agent. Endogenous sources of ethylene are lipid peroxidation reactions and gut microflora [117,118]. N7-(2-hydroxyethyl)guanine (7HEG) modifications account for approximately 95% of DNA adducts due to EO [116]. 7HEG adducts are not promutagenic alone but they lead to depurination of DNA and hence the resulting abasic site can generate mutations during DNA replication. Other minor adducts can also be mutagenic, but while EO has been proved to be genotoxic, the mechanisms of mutagenicity have not been fully described [120].

Metabolism is connected both to the generation of alkylating agents and also to the repair of alkylated bases. Alkylation adducts can be removed by the dealkylases AlkB homologs 2 and 3 (ALKBH2/3), which use α-ketoglutarate (α-KG) as a key substrate. α-KG is produced by the oxidative decarboxylation of isocitrate, catalyzed by isocitrate dehydrogenase (IDH). Many tumor cells carry a mutation in *IDH*, which confers a neomorphic activity on the encoded enzyme and results in the production of the oncogenic metabolite 2-hydroxyglutarate (2HG) from α-KG [138]. Moreover, 2HG can also be generated by lactate dehydrogenase following hypoxia [139]. ALKBH2 and ALKBH3 are inhibited by 2HG and therefore tumor cells with mutations in *IDH* are defective in repairing AlkB substrates, such as 1meA [140]. Similarly, glutamine deficiency induces a decrease of α-KG production, which reduces the activity of AlkB homolog enzymes and prevents the repair of DNA alkylation damage [141]. These findings support an interplay between metabolism and the DDR, especially in cancers, through both the generation of DNA damage and the regulation of DNA repair by metabolites.

## 4. Alterations in dNTP Pools Generate DNA Damage

A correct balance of deoxynucleoside triphosphate (dNTP) levels is crucial for optimal DNA replication and thus for maintaining genome stability. Two distinct pathways produce dNTPs, the de novo biosynthesis pathway and the salvage pathway [142]. The main enzyme in de novo dNTP synthesis is the ribonucleotide reductase (RNR), which reduces all four ribonucleoside diphosphates (rNDP) to the respective deoxynucleoside diphosphates (dNDP), which are then phosphorylated to dNTP. The nucleotide salvage pathway recovers bases and nucleosides formed by the degradation of RNA and DNA, for instance through the action of deaminases, phosphohydrolases, and the triphosphohydrolase SAMHD1 (sterile α motif (SAM) and histidine/aspartate (HD)-domain containing protein 1) [143]. dNTP production in healthy cells is coupled with DNA synthesis and therefore expression of RNR and SAMHD1 is regulated throughout the cell cycle. Their activities are also allosterically controlled by dNTP concentrations [144,145].

Many metabolites and metabolic reactions are involved in nucleotide synthesis. For example, the PPP, with the rate-limiting enzyme G6PD, generates ribose 5-phosphate. In addition, glucose and glutamine feed into both purine and pyrimidine metabolism to donate carbons and nitrogens. Coordinated degradation of dNTPs is also critical for genome maintenance, and, as such, catabolic enzymes are able to limit the expansion of dNTP pools. dNTP pool alterations contribute to replication stress, enhanced mutagenesis, and genomic instability. Thus, the balance of dNTPs is found to be altered in cancer cells [142,146].

Mutations within proto-oncogenes induce aberrant regulation of the cell cycle and proliferation. This can lead to uncoupling of nucleotide synthesis and DNA replication, thus resulting in depletion of nucleotide pools and replication stress. For instance, activation of the Rb-E2F pathway, which controls S phase entry, causes replication stress and chromosomal instability, driving oncogene-induced transformation [147]. Normally, the accumulation of DNA damage due to dNTP depletion would lead to oncogene-induced senescence (OIS) [148], however additional mutations in genes involved in the senescence pathway, such as *TP53*, as well as increased expression of genes involved in nucleotide production mediated by the transcription factor c-Myc can increase the dNTP pools, thus reducing genome instability and allowing cells to escape senescence [147,149,150]. dNTP depletion, DNA damage, and replication stress are thus initial steps in cellular transformation. Next, precancerous cells undergo metabolic adaptation to sustain proliferation by, notably, synthesizing more dNTPs, which further drives tumorigenesis.

Quantification of intracellular dNTP concentrations has revealed increased nucleotide pools in transformed cells [151]. This might be due to a diversion of metabolites into nucleotide synthesis pathways. In support of this, argininosuccinate synthase (ASS1), which catalyzes the penultimate step in the synthesis of arginine and is involved in the urea cycle, is frequently silenced in multiple cancers. Decreased activity of ASS1 enhances availability of its substrate—aspartate—which can then be used for the synthesis of pyrimidine nucleotides [152]. Furthermore, a metabolic link between the enzymes that function in the urea cycle and pyrimidine synthesis has implicated mutations in the oncogene *KRAS* and the tumor suppressor *STK11* in non-small-cell lung cancer, leading to aggressive tumors. Cells with such mutations accumulate urea cycle metabolites and overexpress carbamoyl phosphate synthetase 1 (CPS1), which initiates nitrogen disposal. CPS1 enables an unconventional pathway of nitrogen flow from ammonia to pyrimidine, by providing an alternative pool of carbamoyl phosphate which can be used in pyrimidine biosynthesis [153]. One last example of the rewiring of metabolites in cancer cells for nucleotide production is represented by alterations in glutamate. Cells derived from glioblastoma convert glutamate into glutamine upon glutamine starvation, using glutamine synthetase (GS). Instead of supporting anabolism through the TCA cycle via glutaminolysis, glutamine carbons are used to fuel de novo purine biosynthesis. GS can either be overexpressed in cells derived from glioblastoma directly, or glutamine can be secreted by surrounding astrocytes—which have high GS expression—and be imported by the glioblastoma cells [154].

Enhanced dNTP production in cancer can also occur through regulation of enzymes in the dNTP synthesis pathways. For instance, lack of ATM in *TP53* WT cells bypasses senescence induced by replication stress, which normally suppresses transformation. Indeed, p53 normally inhibits G6PD, but ATM inactivation abrogates this inhibition, thus upregulating the PPP through increased G6PD activity and enhanced glucose and glutamine consumption, which in turn restores dNTP levels. This is contrary to the function of ATM in the activation of G6PD through phosphorylation of Hsp27 (see Section 2.2). Thus, ATM status couples replication stress and metabolic reprogramming during senescence [155]. On the contrary, inhibition of the replication stress sensing kinase ataxia telangiectasia and Rad3-related protein (ATR) in leukemia cells down-regulates the activity of RNR and deoxycytidine kinase (dCK), rate-limiting enzymes of respectively the de novo and salvage dNTP synthesis pathways [57]. Targeting the remaining activities of RNR and dCK by specific inhibitors triggers lethal replication stress in vitro, suggesting a therapeutic opportunity for leukemia and potentially other cancers.

Additionally, post-translational modifications of proteins in cancer cells can also promote nucleotide synthesis. G6PD is glycosylated upon hypoxia—which often occurs in tumors—which leads to enhanced activity, thus increasing glucose flux through the PPP. Among other consequences, it provides more nucleotide precursors, sustaining the proliferation of cancer cells [156]. Finally, the key enzyme involved in dNTP catabolism, SAMHD1, is often depleted or mutated in tumors. Overexpression of SAMHD1 in a lung adenocarcinoma cell line leads to dNTP depletion and reduces cellular proliferation [157]. While adaptation to supply more dNTPs sustains the active proliferation of cancer cells, it is also a source of genomic instability, which can further drive tumorigenesis. Indeed, an excess of dNTPs increases mismatches, due to the formation of non-Watson–Crick base pairs. Moreover, it also inhibits proofreading, a phenomenon known as the “next-nucleotide effect”, because it forces the replicative DNA chain elongation past the site of a mismatch before the polymerase can correct the error. Hence, an increased dNTP pool is linked with genomic instability and a mutator phenotype [158,159,160].

Finally, imbalances in the nucleotide pools are also responsible for the generation of DNA damage. For instance, the one-carbon pathway is crucial for maintaining a balanced pool of nucleotides. Folate and other methyl donors supplied by the diet are required for transferring 1-carbon units in the de novo synthesis of nucleotides. Folate deficiency decreases synthesis of dTTP and purines, while dUTP and dCTP levels are not affected. An imbalance in the dCTP/dTTP ratio causes spontaneous mutagenesis, by misincorporation of the nucleotide in excess (next nucleotide effect) [161,162]. Moreover, an increased dUTP/dTTP ratio, due to deficient methylation of dUMP to dTMP, leads to incorporation of uracil during DNA replication, because DNA polymerases are not able to distinguish dUTP from dTTP. While substitution of dTTP by dUTP does not affect Watson–Crick base pairing, it does impact on genome stability, since uracil-DNA glycosylase and apyrimidinic endonuclease efficiently excise dUTP from the DNA, creating promutagenic abasic sites and transient SSBs [163]. Similarly, it has been reported that concomitant increased expression of the RNR subunit R2, resulting in enhanced expression of dUTP, and decreased expression of dUTPases, which impairs dUTP sanitization, lead to a poor prognosis in colorectal and breast cancers [164]. Finally, it has also been shown that dihydropyrimidines, degradation products of pyrimidines, interfere with DNA replication. These cytotoxic metabolites are produced by dihydropyrimidine dehydrogenase (DPD) from pyrimidines and are further degraded by dihydropyrimidinase (DHP). In untransformed cells, the activity of DHP is essentially restricted to the liver and the kidneys, but cancer cells display altered pyrimidine catabolism, with increased activity of DPD and DHP. The accumulation of dihydropyrimidines induces the formation of DNA–protein crosslinks and generates replication and transcription stress, and thus depletion of DHP impairs the proliferation of cancer cells, suggesting a target for cancer therapy [165,166]. Free nucleotides can additionally be damaged, for example by oxidation, and their incorporation into DNA during replication can then lead to mutations. Sanitizing enzymes, such as Mut-T homolog 1 (MTH1), which converts 8-oxoGTP to 8-oxoGMP, help to maintain healthy dNTP pools. Cancer cells have a strong dependency on MTH1 activity, due to increased ROS production and oxidized dNTPs, hence suggesting a therapeutic opportunity [167,168].

## 5. Metabolism is Involved in the Repair of DNA Double-Strand Breaks

DSBs are highly toxic lesions, which, if left unrepaired, lead to replication blocks and cell death. Cells have thus evolved efficient repair mechanisms to deal with this genomic threat, and recent studies show that metabolism is directly involved in these processes (Figure 2). DSBs can mainly be repaired by two main pathways: error-free HR that requires end-resection and a repair template and thus functions in the G2/S phase of the cell cycle, and error-prone non-homologous end-joining (NHEJ) that directly ligates the DSB ends, generating small insertions and deletions (indels).

### 5.1. Regulation of dNTP Pools is Critical for Efficient Repair of DSBs

dNTP-pool alterations can generate DNA damage by creating replication stress, as discussed in Section 4, but they can also prevent the efficient repair of DSBs. Recent findings indicate a necessity to generate dNTPs in the vicinity of DNA damage sites, for example by recruitment of RNR or thymidylate kinase (TMPK) [169,170,171]. Although the number of dNTPs required for DNA repair is small, their concentration is critical, and one hypothesis is that dNTP diffusion could be limiting for DNA synthesis during repair. Moreover, it has been reported that de novo nucleotide production is an important determinant in DSB repair pathway choice. As such, HR, which requires extensive DNA synthesis, depends on dNTP availability to a larger extent than NHEJ [172].

Metabolic reactions play a critical role in the synthesis of a balanced pool of nucleotides in response to DNA damage. It has been shown that DNA damage causes cells to rewire their metabolism towards the PPP, to produce ribose 5-phosphate that is used in the synthesis of nucleotides. Activation of ATM, which regulates the response to DSBs, induces G6PD to increase the production of dNTPs and promote DNA repair [30]. Similarly, it has been reported that GS is overexpressed in radioresistant cancer cells. This leads to a decrease in glycolysis and glutamine catabolism, causing an increased production of glutamine that can be used by cells for nucleotide precursor synthesis. This cellular metabolic flux reprogramming allows for better DNA repair capacities, resistance to radiotherapy, and a survival advantage [173].

The phosphoglycerate mutase 1 (PGAM1) additionally functions in DSB repair, and more specifically in HR, through regulation of the dNTP pools. PGAM1 converts 3-phosphoglycerate (3PG) into 2-phosphoglycerate (2PG) and coordinates glycolysis and the PPP. PGAM1 inhibition causes dNTP depletion, through accumulation of 3PG, which inhibits 6-phosphogluconate dehydrogenase (6PGD) in the PPP. The resulting replication stress response leads to the activation of the APC/C-Cdh1 E3 ubiquitin ligase, which then ubiquitinates the CTBP-interacting protein (CtIP), leading to its degradation. Since CtIP is a critical component of DNA end-resection in HR, this repair pathway is thus inhibited [174].

Finally, the glycolytic enzyme 6-Phosphofructo-2-Kinase/Fructose-2,6-Biphosphatase 3 (PFKB3) is also involved in HR, independently from its metabolic function. This isoform, normally induced by hypoxia, extracellular acidosis, or inflammation, is overexpressed in cancer cells. After DSBs induced by irradiation, PFKB3 relocalizes into nuclear foci in an ATM-γH2AX-MDC1 dependent-manner. It then mobilizes the RRM2 subunit of RNR, probably to generate a local pool of dNTPs at the site of DNA damage, hence supporting DNA synthesis during DSB repair. Inhibition of PFKB3 could thus provide cancer-specific radiosensitization [175].

### 5.2. Metabolic Regulation of Epigenetic Marks Influences DSB Repair

Chromatin state, regulated by epigenetic marks, is of crucial importance in the recruitment of DSB repair factors and in the choice of repair pathways. The availability of methyl- and acetyl-group donors is strongly regulated by metabolic reactions, illustrating an additional layer of complexity in the crosstalk between metabolism and DNA repair.

ATP-citrate lyase (ACLY), which regulates the availability of acetyl-CoA by catalyzing the conversion of oxidation-derived mitochondrial citrate into acetyl-CoA and oxaloacetate, is upregulated in several cancers [176]. ACLY can be localized in the nucleus, especially in the S/G2 phase of the cell cycle, where it functions to provide acetyl-CoA in the vicinity of a DSB. Moreover, it is phosphorylated downstream of ATM and AKT activation, following IR, which enhances its activity. Histone acetylation close to DSB sites facilitates BRCA1 recruitment, while interfering with the binding of the NHEJ factor 53BP1, thus promoting faithful repair by HR [177,178,179].

Another metabolic alteration common in different cancers is the increased production of the oncometabolite 2HG, which inhibits α-KG dioxygenases [180], such as the Alkb homolog dealkylases (see Section 3), as well as the lysine-specific demethylases KDM4A and KDM4B. KDM4B inhibition leads to basal hypermethylation of H3K9, masking a local spike in trimethylation at DSB sites, which impairs recruitment of repair factors, such as TIP60 and ATM. End-resection and recruitment of downstream repair factors is consequently impaired [181]. Thus, the epigenetic changes induced by 2HG generate an HR defect, which confers a “BRCAness” phenotype, hence sensitizing such tumors to PARP inhibitors. This oncometabolite is, as such, a double-edge sword, which both drives carcinogenicity and creates a cellular vulnerability to chemotherapy.

The metabolites fumarate and succinate are also important in the regulation of epigenetic marks following DSBs. They can accumulate in cancer cells due to mutations in fumarate hydratase (FH) and succinate dehydrogenase (SDH), for example in Krebs cycle-deficient hereditary cancer syndromes, thereby connecting metabolism to the DDR in cancers. Similar to 2HG, they inhibit α-KG dioxygenases, such as KDM4A/B, increasing basal level of H3K9me3, and lead to a defect in HR [182,183]. Moreover, fumarate accumulation, in the context of FH deficiency, leads to a bypass of checkpoint maintenance, thus inducing endogenous DNA damage and increasing resistance to IR [184]. However, local production of fumarate by FH, which catalyzes the reversible reaction of fumarate to malate, has also been shown to enhance NHEJ. FH translocates to the nucleus upon DNA damage, where it is phosphorylated by the catalytic subunit of DNA-PK (DNA-PKcs). This phosphorylation leads to the interaction of FH with the histone H2A.Z at DSB sites, where the enzyme can locally produce fumarate. Fumarate then inhibits another (α-KG)-dependent lysine demethylase, KDM2B, and the resulting persistence of dimethylated histone H3 at lysine 36 facilitates the binding of more DNA-PK complex and other pro-NHEJ proteins [185,186].

Finally, *N*-acetyl-glucosamine (GlcNAc) production by the hexosamine biosynthetic pathway and resulting protein O-GlcNAcylation is elevated in cancer cells due to increased glucose and glutamine metabolism. This post-translational modification can drive DSB repair and resistance to therapy-induced senescence. The enhancer of zeste homolog 2 (EZH2), a histone-lysine *N*-methyltransferase, is stabilized and activated by O-GlcNAcylation. Its activity then promotes DSB repair by enhancing H2K27 trimethylation, an important determinant in NHEJ repair. Metabolic rewiring thus protects cancer cells against radiotherapy, increasing DNA repair and limiting senescence [187].

### 5.3. Metabolic Enzymes and Metabolites are Directly Involved in DSB Repair

Metabolic enzymes and metabolites have been reported to be directly involved in DSB repair, independent from their metabolic functions. The glycolytic enzyme pyruvate kinase M2 (PKM2) isoform is highly expressed in tumors. PKM2 localizes in both the cytoplasm and nucleus and has multiple implications in cancer development, driving the Warburg effect, activating the transcription of oncogenes but also promoting HR. After DSB generation, ATM phosphorylates PKM2, which is then retained in the nucleus and activates CtIP by phosphorylation to increase HR [188]. Moreover, PKM2 directly phosphorylates H2AX following DNA damage, generating γ-H2AX. Hence, it modulates genomic instability and thus plays an important role in tumorigenesis [189].

Next, polyamines are often elevated in tumors, due to the upregulation of ornithine decarboxylase (ODC), the rate-limiting enzyme in their biosynthesis. These metabolites promote cell growth and proliferation and are also directly involved in DSB repair. It has been reported that depletion of polyamines prevents DNA repair, and that the unrepaired DSB then induces apoptosis of proliferating cells. Polyamines specifically function in HR, where they stimulate RAD51-mediated strand exchange by modifying the properties of DNA and facilitating the capture of duplex DNA by the RAD51 presynaptic filament [190]. Moreover, depletion of polyamines by inhibition of ODC sensitizes triple-negative breast cancer cells to cytotoxic chemotherapies, thus indicating a targetable metabolic vulnerability [191].

Finally, the cofactor NAD^+^ (see Section 2) is involved in the response to DNA DSBs. The inhibition of NAMPT and NAPRT sensitizes cancer cells to DNA damage, such as DSB-inducing drugs or alkylating agents. This sensitization is more pronounced in cancers that are HR-deficient [52,192]. Additionally, NAD^+^ depletion decreases NHEJ through impairment of SIRT1, an NAD^+^-dependent deacetylase that enhances DNA repair capacity through the deacetylation of Ku70, a key component of NHEJ [193]. Moreover, it has been reported that NAMPT physically associates with key factors of HR and NHEJ (CtIP and DNA-PKcs/Ku80), suggesting a direct role for this metabolic enzyme in DSB repair [194]. Inhibitors of NAMPT are currently used in the clinic for the treatment of cancers, in combination with genotoxic drugs [33].

It should be mentioned that DSBs can be generated in a programmed manner. For example, generation of DSBs and their repair by NHEJ are necessary for the development of lymphocytes through V(D)J recombination and class-switch recombination, to increase antibody diversity [195]. Metabolism can also regulate these programmed DNA lesions and thus the development of lymphocytes. For example, the metabolic sensor AMP-activated protein kinase (AMPK), which is activated in response to metabolic stress and energy depletion, directly phosphorylates the recombination-activating gene 1 protein (RAG1), which is part of the endonuclease complex catalyzing DNA cleavage during V(D)J recombination [196].

## 6. Outlook: Approaches to Identify Interactions between Metabolism and the DNA Damage Response

The interplay between metabolism and the DNA damage response is indisputable and is even more pronounced in cancer cells, as illustrated throughout this review. Yet, we propose that many interactions between these two fundamental pathways remain unknown, and a better comprehension of this interplay is crucial to understand resistance mechanisms to genotoxic therapies and to design improved treatments for patients. The development of high-throughput technologies, such as genetic and chemical screens as well as mass spectrometry-based approaches, will enable a more comprehensive mapping and understanding of the interactions between metabolism and DDR pathways in the near future (Figure 3).

Chemical screens have been used for decades to assess the toxicity of drugs, and the development of high-throughput readouts now allow for the analysis of bigger libraries comprising hundreds to thousands of compounds [197,198]. For instance, screening of a chemical library allowed the identification of a potent inhibitor of a mutated form of isocitrate dehydrogenase 2 (IDH2), which is expressed in acute myelogenous leukemia, and which is now used as a chemotherapeutic [199]. Compounds can also be investigated in DNA repair-deficient cell lines, to identify synthetic lethal interactions [200]. Chemical screens can furthermore lead to the discovery of new gene functions, or new gene–gene interactions, hence contributing to basic, fundamental knowledge [201]. To specifically study crosstalk between metabolism and DDR, chemical libraries targeting metabolic pathways can be compiled from FDA-approved drugs or drugs in clinical trials, as well as using metabolites and food carcinogens as perturbations. Such screens coupled with next generation sequencing (NGS)-based readout can also inform on mutagenic outcomes of metabolic compounds/inhibitors. These chemical libraries can be designed based on available databases inventorying drugs, metabolites, and toxic compounds [202,203,204], or based on computer-aided molecular design [205,206].

Next to chemical screens, genetic screens are a powerful tool to identify novel gene functions. At first restricted to the use of small interference RNA (siRNA), short hairpin RNA (shRNA) and insertional mutagenesis, recent years have seen a rapid development of the toolbox available for such screens, especially with the advent of CRISPR-Cas9 [207,208,209]. CRISPR screens can be performed both in a pooled manner, where NGS is used to identify enrichment or depletion of sgRNAs, or in an arrayed manner. Such screens have been successfully performed to unravel new genes involved in both DNA repair pathways [210,211] and metabolic processes [212].

Both genetic and chemical screens are very powerful approaches, especially when they are combined with a precise readout, tailored to the initial hypothesis, to identify candidates giving rise to a particular phenotype. The simplest readout is to record the effect of chemical or genetic perturbations on cell survival, by looking at depleted cell populations (i.e., negative selection) or enriched cell populations (i.e., positive selection). Readouts based on survival can be used for both pooled and arrayed screens. Positive selection is typically used to study resistance mechanisms in cancer cells since it allows for the identification of perturbations that confer a growth advantage after treatment with a drug. This approach has, for example, been used to identify genes whose loss is involved in resistance to vemurafenib, a therapeutic RAF inhibitor for melanoma, in a genome-wide CRISPR-Cas9 knockout screen [213]. On the other hand, screens based on negative selection are mostly used to determinate genes essential for cell survival in certain conditions and can allow the identification of cellular vulnerabilities that are specific to cancer cells [214].

Readouts based on more complex and specific phenotypic measurements can also be used. They can be based on fluorescent reporters or on staining of endogenous proteins. Fluorescent activated cell sorting (FACS) or high-throughput microscopy can then be utilized to measure the response of the cells to the genetic or chemical perturbations. Until recently, microscopic readouts were restricted to arrayed screens, but new technologies have adapted this approach to make it amenable to pooled screens as well [215]. To date, multiple optical readouts have been established to study DNA damage and repair. For the detection of DNA damage, one of the most versatile approaches is the staining of γ-H2AX, as a generic marker of DNA damage [216]. Other optical tools have been developed to detect particular types of damage, such as luminescent probes binding specifically to mismatches or abasic sites [217]. The detection sensitivity of such probes will need to be optimized for use in high-throughput experiments. The direct assessment of specific DNA repair pathways can also give more precise insights into how a specific perturbation impacts the DDR. This can be done both by staining of DNA repair proteins unique to a given repair pathways or by using DNA repair assays based on fluorescent probes. For example, quantification of RAD51 foci after IR in a genome-wide siRNA screen identified a role in homologous recombination for the glycolytic enzyme PFKB3 [175,218]. The use of plasmid reporters for different types of DNA damage, which are either fluorescent when damaged or after repair, can allow for the measurement of different DNA repair activities in combination [219,220].

With the increasing development of screening approaches, classical methods to study the DNA damage response have been adapted to ensure their compatibility with high-throughput formats. An example is the slide-based comet assay technique, which has recently been optimized to be performed in a 96-well platform [221,222]. Another readout that has been adapted for pooled genetic screens is transcriptome profiling. This readout is extremely informative since it is data-driven and hence allows for assessing modifications of many regulatory pathways simultaneously [223,224]. Considering the promises of chemical and genetic screens for deciphering cellular mechanisms, more and more readouts will undoubtedly be optimized for high-throughput formats.

Finally, mass spectrometry-based technologies offer another high-throughput approach with the potential to uncover new interactions between metabolism and the DDR. Indeed, different proteomic techniques can be used to study DNA repair mechanisms. Firstly, changes in abundance of proteins in cells treated with a DNA damaging agent compared to untreated cells, or more specifically in abundance of post-translational modifications regulating the DDR, give particular insights into activated pathways and proteins that function in DNA repair [225,226]. Also, identification of protein–protein interactions following DNA damage is a way to discover new components of the DDR [227]. Other methods have been developed to specifically investigate the chromatin composition after DNA damage, such as CHROMASS (chromatin mass spectrometry) [228]. This technique has allowed for investigation of the kinetics by which DNA repair factors are recruited to interstrand crosslinks. Similarly, there is an ever-increasing development of metabolomics, including an improvement in the number of metabolites that can be precisely measured simultaneously. Metabolomics can be instrumental in allowing for the identification of metabolites that modulate phenotypes, such as the oncogenic metabolite 2HG [138].

## 7. Conclusions

A wealth of studies has uncovered a dynamic interplay between metabolic factors and the DDR. Metabolism is involved in both the generation and repair of DNA damage. This is particularly relevant in cancer, which can be considered both a metabolic and a genetic disease. The development of high-throughput technologies will undoubtedly assist in answering critical research questions that will help us understand how metabolic reprogramming is involved in cancer therapy resistance and how this knowledge can be used to develop combination therapies targeting both metabolism and DNA damage.

## Figures and Tables

**Figure 1 cancers-12-02051-f001:**
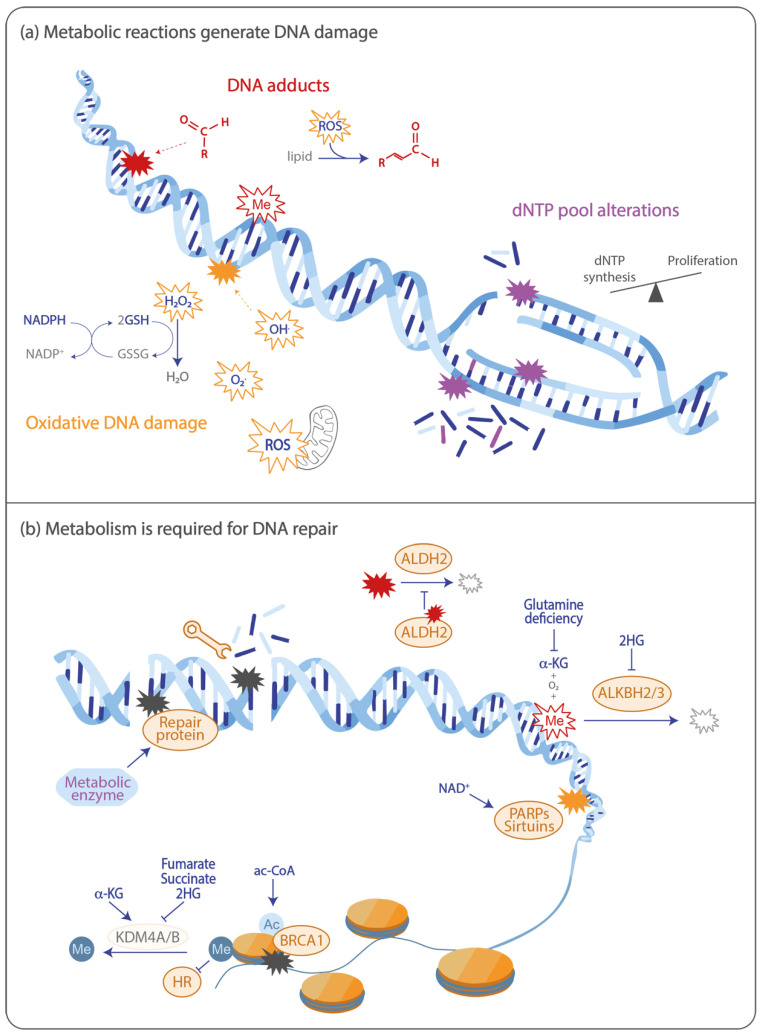
Interplay between cellular metabolism and the DNA damage response in cancer. (**a**) Metabolic reactions generate DNA damage. Toxic metabolites, such as alkylating agents and aldehydes, which are produced by lipid peroxidation and other metabolic pathways, can generate DNA adducts. Reactive oxygen species (ROS), mainly produced by oxidative phosphorylation, generate oxidative DNA damage. ROS can be detoxified by antioxidants, such as glutathione (GSH) and nicotinamide adenine dinucleotide phosphate (NADPH). The increased proliferation of cancer cells (as depicted in the top strand) can deplete dNTP pools causing replication stress and, consequently, DNA double-strand breaks (DSBs). Imbalances in nucleotide pools (as depicted in the bottom strand) cause misincorporations during replication, which can lead to mutations. (**b**) Metabolism is required for DNA repair. Metabolic enzymes recruit DNA repair proteins to the sites of DSBs, and the production of a local pool of nucleotides in the vicinity of DSBs is needed for accurate repair. Proteins involved in DNA adduct repair are regulated by metabolites. The dealkylases AlkB homologs 2 and 3 (ALKBH2/3) use α-ketoglutaric acid (α-KG)—produced from glutamine—as a key substrate and are inhibited by the oncometabolite 2-hydroxyglutarate (2HG). The enzyme aldehyde dehydrogenase 2 (ALDH2), which is involved in the degradation of aldehydes, can itself be crosslinked by this metabolite, hence reducing its activity. Poly-ADP-ribose polymerases (PARPs) and sirtuins are essential enzymes in the DNA damage response (DDR) and use the metabolite nicotinamide adenine dinucleotide (NAD^+^) as a substrate. Finally, chromatin remodeling and epigenetic marks play a crucial role in DNA repair. The production of acetyl-CoA in the vicinity of DSBs facilitates histone acetylation and BRCA1 recruitment, promoting homologous recombination (HR). On the contrary, the inhibition of the lysine specific demethylases 4A and 4B (KDM4A/B) by 2HG, fumarate, or succinate prevents histone demethylation and subsequent recruitment of HR factors.

**Figure 2 cancers-12-02051-f002:**
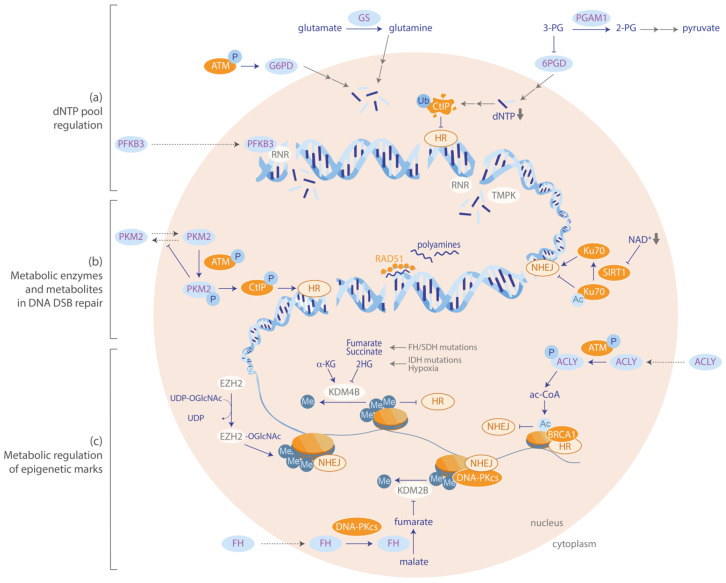
Metabolism and the repair of DNA double-strand breaks. Crosstalk between metabolism and DNA repair can be classified into three categories: (**a**) dNTP-pool regulation is critical for DNA repair, (**b**) metabolic enzymes and metabolites are directly involved in DSB repair, and (**c**) metabolic regulation of epigenetic marks influences DNA repair.

**Figure 3 cancers-12-02051-f003:**
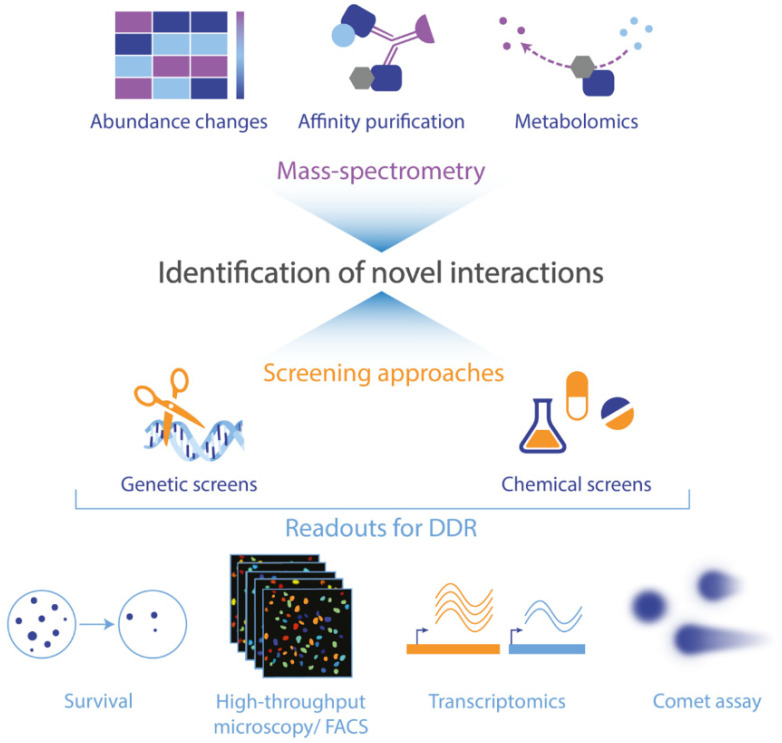
Overview of currently available high-throughput technologies and approaches to discover new crosstalk between metabolism and the DNA damage response (DDR). Proteomics can be used to either quantify the abundance of proteins or post-translational modifications after DNA damage, whereas affinity purification of known DNA repair proteins can identify unknown interactors. Metabolomics can identify metabolites that modulate DNA repair. Genetic or chemical screens utilizing a variety of different readouts, described in more details in the main text, have been developed and represent a useful tool to pinpoint complex interaction networks between metabolism and DDR.

**Table 1 cancers-12-02051-t001:** Endogenous metabolites generate mutagenic DNA adducts. This table indicates the most studied endogenous aldehydes and alkylating agents that generate DNA adducts, summarizing the types of adducts that are created, the endogenous pathways producing these metabolites, and the long-term consequences of the adducts on genome integrity. A detailed overview of the different mutations and impacts on genome instability of these metabolites can be found in more focused reviews [57,58,59,60].

Metabolite	Main DNA Adducts and/or Crosslinks	Pathways Producing the Metabolite	Predicted Impacts on Genome Integrity
Formaldehyde	N2-hydroxymethyl-deoxyguanosine (N2- HOMe-dG) N6-hydroxymethyl-deoxyadenosine (N6-HOMe-dA) N4-hydroxymethyl-deoxycytosine (N4-HOMe-dC) [61] DNA–protein crosslinks [62] DNA intra and interstrand crosslinks [63]	Byproduct of enzymatic oxidative demethylation reactions [64] Methylamine metabolism [65] Myeloperoxidation [66] Methanol metabolism [67]	Base substitutions Frameshift mutations DNA breaks and chromosomal aberrations [68] Tandem bases substitutions
Acetaldehyde	N2-ethylidene-deoxyguanosine (reduced form: N2-ethyl-2′-deoxyguanosine) [69,70] DNA–protein crosslinks [71] DNA intra and interstrand crosslinks [70,72]	Ethanol metabolism [73] Pyruvate, threonine and other metabolic processes [74]	Base substitutions Frameshift mutations DNA breaks and rearrangements [73] Tandem bases substitutions [72]
Methylglyoxal	N2-(1-carboxyethyl)-2′-deoxyguanosine (CEdG) [75,76] Glycation of histones [77] Interstrand crosslinks [78] DNA–protein crosslinks [79]	Side product of glycolysis (Triosephosphate degradation) [80] Product of the degradation of acetone, aminoacetone and threonine [81] Degradation of glycated proteins [82] Lipid peroxidation [83]	Depurination of DNA: promutagenic Replication block: strand breaks, deletions Frameshift mutations
Malondialdehyde (MDA)	Pyrimido [1,2-α]purine-10(3H)-one-2′-deoxyribose (M1dG: main product) [84] N6-(3-oxoprenyl)deoxyadenosine (OPdA) [85] N4-(3-oxoprenyl)deoxycytidine (OPdC) [86] DNA interstrand crosslinks [87] DNA–protein crosslinks	Lipid peroxidation [88] Biosynthesis of prostaglandins [89,90]	Base substitutions [91] Frameshift mutations
4-hydroxy-2-nonenal (HNE)	Substituted 1,N2-propano-2′-deoxyguanosine (4 diastereomers) [92,93]	Lipid peroxidation [60,94]	Base substitutions
2,3-epoxy-4-hydroxynonanal (HNE epoxyde)	Etheno adducts: 1,N2-ethenodeoxyguanosine [95] 3,N4-ethenodeoxycytidine [96] 1,N6-ethenodeoxyadenosine [96]	Oxidation of HNE [60,97]	Base substitutions DNA replication blocade: by-pass by error-prone TLS polymerases [98]
Crotonaledyde (or 2 acetaldehydes)	8-hydroxy-6-methyl-1,N2-propano-2′-deoxyguanosine (2 diastereomers) [99,100] DNA interstrand crosslinks [101] Protein–DNA crosslinks	Lipid peroxidation [60] Metabolite of N-nitrosopyrrolidine [102]	Base substitutions [103] Frameshift mutations [103]
Acrolein	γ-hydroxy-1,N2-propano-2′-deoxyguanosine (γ-OH-PdG) α-hydroxy-1,N2-propano-2′-deoxyguanosine (α-OH-PdG) [104,105] DNA intra and interstrand crosslinks [101,106] Protein–DNA crosslinks	Lipid peroxidation [60,107] Myeloperoxidation in neutrophils and monocytes [108]	Base substitutions [109] Frameshift mutations Tandem bases substitutions [106]
S-Adenosyl methionine (SAM)	N7-methylguanine (7meG) N1- and N3- methyladenine (1meA and 3meA) O6-methylguanine (O6meG) O4-methylthymine (O4meT) O4-ethylthymine [110,111,112]	Synthesized from ATP and methionine [113]	7meG: Harmless but can become an abasic site, promutagenic 3meA: DNA replication block, by-pass by error-prone TLS polymerase O-adducts: Bases mispairing [114]
Ethylene oxide (EO)	N7-(2-hydroxyethyl)dG (main product) N3-(2-hydroxyethyl)dA N3-(2-hydroxyethyl)dU O6-(2-hydroxyethyl)dG [115,116]	Lipid peroxidation [60,117] Gut microflora [118]	Mutagenic (unclear mode of action) [119,120]

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
