# Peer review of "Interplay between Cellular Metabolism and the DNA Damage Response in Cancer"

_cancers, 2020, doi:10.3390/cancers12082051_

Round 1

Reviewer 1 Report

the authors' work is a very interesting contribution and an overview of the relationship between metabolism and DNA damage response in cancer knowledge. the discussion is very useful to stimulate future experimental research on the topic and in other aspects such as cancer therapy and prevention.

Author Response

We appreciate the Reviewer’s positive feedback and we are very pleased that our review may stimulate further research on the crosstalk between metabolism and the DNA damage response.

Reviewer 2 Report

I have read this extensive review on the interplay between DNA damage responses and different aspects of metabolism with great interest. I am impressed both by the degree of overview and the level of detail. It is very unusual during a referee task to be able to say that I do not actually have any complaints, but is very happy with this mansucript in its present form. I am sure there are some typos and sentences that could be improved but I did not even find that.

Author Response

We thank the reviewer for the positive comments.

Reviewer 3 Report

The authors performed a very extensive and complete bibliographic review on the dynamic interplay between metabolic factors and DNA Damage Repair mechanisms with focus in cancer. The manuscript is very well written and organized. The figures included in the article help the reader to structure and compile the large amount of scientific information available on these topics.

Author Response

We thank the reviewer for the positive review of our manuscript.